# Protein and Carbohydrate Fractionation to Evaluate Perennial Ryegrass (*Lolium perenne* L.) Accessions

Martin Gierus *,† , Heba Sabry Attia Salama ‡ , Marc Lösche § , Antje Herrmann ‖ and Friedhelm Taube

Grass and Forage Science/Organic Agriculture Group, Institute of Crop Science and Plant Breeding, University of Kiel, 24118 Kiel, Germany; heba.salama@alexu.edu.eg (H.S.A.S.); loeschem@gmx.de (M.L.); antje.herrmann@llh.hessen.de (A.H.); ftaube@gfo.uni-kiel.de (F.T.)

* Correspondence: martin.gierus@boku.ac.at
† Present address: Institute of Animal Nutrition, Livestock Products and Nutrition Physiology (TTE), Department of Agrobiotechnology, IFA-Tulln, University of Natural Resources and Life Sciences, 1190 Vienna, Austria.
‡ Present address: Crop Science Department, Faculty of Agriculture, Alexandria University, Aflaton St., El-Shatby, Alexandria 11865, Egypt.
§ Present address: Bundesamt für Verbraucherschutz und Lebensmittelsicherheit, Bundesallee 51, 38116 Braunschweig, Germany.
‖ Present address: Landesbetrieb Landwirtschaft Hessen, 36251 Bad Hersfeld, Germany.

**Abstract:** Precise information about carbohydrates and proteins in relation to their utilization in the rumen is useful for the breeding purposes of perennial ryegrass cultivars used for animal nutrition. The objective of the current study was to evaluate 20 diploid perennial ryegrass accessions from the intermediary heading stage. The ruminal kinetics of different carbohydrate and protein fractions of grasses and legumes are important for forage breeding programs. The Cornell Net Carbohydrate and Protein System (CNCPS) was used to provide such information. Accession-based variation can be observed after considering dynamic degradation processes. Variation among the 20 accessions was observed. Ruminally digested (RDC) and undigested (UDC) carbohydrate and ruminally degraded (RDP) and undegraded (UDP) protein contents, total carbohydrate and total protein contents, and carbohydrate and protein fractions for the first cut and annual averages show significant differences. Although the variation was smaller for the protein fractions, the calculated usable protein content (uCP, sum of UDP and amount of synthesized microbial protein in the rumen) in the small intestine is mainly influenced by decreasing the neutral detergent fiber (NDF) and increasing the sugar content among cultivars. Carbohydrate and protein fractionation is suitable for characterizing perennial grass accessions as it uses parameters relevant to ruminant nutrition, allowing a step forward in forage plant breeding for forage quality. To conclude, using parameters related to ruminal degradation kinetics should favor the selection of accessions with higher amounts of ruminally digested carbohydrates (RDC). The selection of accessions based on protein quality (proportion of UDP) is less favorable for achieving a higher usable protein content.

**Keywords:** fractionation; rumen kinetics; perennial ryegrass; CNCPS

## 1. Introduction

Perennial ryegrass has high nutritional value and has been considered the most important grassland species in Europe for quite a long time [1]. It is important to consider that the production of high-quality grass herbage throughout the growing season is critical for cost-effective livestock production in forage-based systems [2]. One of the main problems in intensive European forage-based systems is the incompatibility between the rapid rate of protein degradation and the slow rate of energy release in the rumen [3,4]. After fresh plant material enters the rumen, i.e., the step after ingestion and before extensive plant cell wall degradation by rumen microorganisms, there is often a period of rapid proteolysis that provides more N than is required by microbes in the rumen [5,6]. This asynchrony leads

to ammonia accumulation in the rumen, which is related to high N loss. The ammonia in the rumen is converted to urea in the liver and is mainly excreted in the urine [6,7]. Such asynchrony is one reason for environmental N pollution in soil and groundwater. The poor quality of protein in forages also leads farmers to purchase protein-rich concentrates with higher UDP (undegraded dietary protein) content, adding more (crude protein) CP to rations and resulting in even higher on-farm N surpluses. Increasing the efficiency of nitrogen use in ruminants is therefore a major goal, and understanding protein degradation in forages is essential.

One possible solution is to breed forage plants with a focus on the needs of ruminants while selecting new varieties. Breeding grasses for improved nutritional value would support efficient farming systems. To approach this efficiency, breeding grasses to improve utilization by selecting for higher levels of rapidly degradable carbohydrates has been used [8]. This may result in the improved forage intake and nutrient supply of high-yielding dairy cows. The selection progress could be efficiently evaluated using the Cornell Net Carbohydrate and Protein System (CNCPS), which is used for cattle diet formulation [9]. The CNCPS was first published in 1992 and 1993 in a series of four papers [9–12], but the model has been continuously refined and improved over the past 30 years (e.g., [13–20]). The CNCPS uses kinetic parameters to simulate rumen metabolism not found in other feed evaluation systems, specifically carbohydrate degradation rates, predicted rumen pH, and rumen nitrogen and peptide balance. It also has equations to estimate the rate of fermentation and passage of forage carbohydrate and protein fractions. This information can be used as a basis for predicting metabolizable energy (ME) and the proportion of feed protein digestible in the small intestine [9,12,21,22].

Another way to improve feed quality is to select for protein quality. In this case, forages with different degradation kinetics, i.e., with lower protein degradation rate in the rumen, should be selected, also considering the initial proteolytic rate [5] and the proportion of non-protein N (NPN) already present in the forage. Such parameters of the nutritional value of forages are highly correlated with maturity at harvest, cutting frequency, and mineral N fertilization rate. To meet the requirements of dairy cows, the amount of usable crude protein (uCP) in the duodenum is a resultant parameter in breeding programs to improve the nutritive value of perennial ryegrass closer to the needs of ruminants.

However, forage plant breeders need basic information on key nutritional parameters of accessions that show sufficient variation and can be cultivated to improve animal N use efficiency. Such information, focusing on forage breeding for quality, is scarce. Variation among perennial ryegrass cultivars (only diploid at the intermediate heading) should be large enough to differentiate them to meet requirements in ruminant nutrition and selection of cultivars for breeding purposes. Selection for higher ruminal undegradable protein content in perennial ryegrass (as a result of lower protein degradation rate) supports increasing protein amounts flowing to the duodenum.

Limited breeding progress in the nutritive value of perennial ryegrass has been achieved in recent decades compared to other crops. One of these breeding advances affecting ruminant nutrition has been the introduction of high-sugar genotypes [8,23,24]. It is therefore very important to identify accession-related differences in the nutritional value of perennial ryegrass.

In the present study, we hypothesize that the CNCPS provides sufficiently detailed information to quantify substantial variation among perennial ryegrass accessions. The objectives of the present study were: (I) to analyze and evaluate the composition of the crude protein and carbohydrate fractions of different perennial ryegrass accessions and (II) to provide information on parameters resulting from the rumen kinetics of the tested accessions using CNCPS.

## 2. Materials and Methods

### 2.1. Plant Material, Location, and Experimental Design

The field experiment was conducted for 2 years at 3 sites in northern Germany, namely Asendorf (52°46′ N, 9°01′ E, altitude 38 m a.s.l.), Malchow (53°59′ N, 11°28′ E, altitude 5 m a.s.l.), and Hohenschulen (54°18′ N, 9°58′ E, altitude 24 m a.s.l.). The three sites are characterized by sandy loam soils. Mean annual temperatures in the first year of the experiment were 10.4, 10.4, and 10.0 °C, and in the second year 10.5, 10.7, and 9.9 °C for the three sites Asendorf, Malchow, and Hohenschulen, respectively. The first year was characterized by a low average annual precipitation of 528, 437, and 593 mm in Asendorf, Malchow, and Hohenschulen, respectively. In the second year of the experiment, higher average annual precipitation values of 802, 759, and 773 mm were recorded for the three sites, respectively. The plant material studied consisted of twenty diploid perennial ryegrass (*Lolium perenne* L.) accessions with an intermediate heading. Accessions 9 and 10 are known to be high-sugar accessions. Accessions are referred to in this publication by numbers only, as not all are registered varieties. In addition, the purpose was to obtain a representative sample of varieties (or candidates). Three replicated 3 × 6.5 m plots per accession were sown in a randomized complete block design.

### 2.2. Management and Sampling

The experimental plots at the three sites were sown in September 2005, i.e., the previous year, and sampling was carried out in the following seasons of 2006 and 2007. All plots were treated equally, and the same amount of fertilizer was applied, 300 kg N/ha (in the form of calcium ammonium nitrate) divided into four applications, namely 100, 80, 80, and 40 kg N/ha, in both experiments before the first, second, third, and fourth harvests, respectively. In addition, 80 kg $P_2O_5$/ha was applied as a single dose in the spring of each year. A potassium fertilizer (40% $K_2O$, 6% $MgSO_4$, 4% S, 3% Na) was applied at a rate of 360 kg $K_2O$/ha in two applications of 200 and 160 kg $K_2O$/ha before the first and third harvests, respectively. After the second harvest, Folicur© (1-(4-chlorophenyl)-4,4-dimethyl-3-(1,2,4-triazol-1-ylmethyl)-pentan-3-ol) was sprayed to control crown rust at a rate of 0.7 L/ha two weeks after each harvest. At the time of sampling, plots were harvested with a Haldrup© plot harvester to a stubble height of 5 cm. Representative subsamples were dried to constant weight at 60 °C to determine dry matter (DM) content.

### 2.3. Maturity Stage Determination

The mean stage by count (MSC) was calculated as the average of the individual stage categories present in the herb sample, weighted by the number of tillers in each stage [25]. Briefly, the MSC of plants was quantitatively monitored at each sampling date at the Hohenschulen site by randomly cutting a representative sample of approximately 50 tillers from each plot to ground level and applying the values to the following equation:

$$\text{MSC} = \frac{\sum_i^n S_i \times N_i}{C}$$

where $S_i$ is the stage category, $N_i$ is the number of tillers in $S_i$, and $C$ is the total number of tillers. The MSC values and heading dates are shown in Table 1.

### 2.4. Analytical Procedures

Dried subsamples were uniformly ground to a particle size of 1 mm using a Cyclotec 1093 sample mill (Foss, Hilleroed, Denmark). All available samples were scanned twice using an NIR-Systems 5000 monochromator (Perstrop Analytical Inc., Silver Spring, MD, USA) with software (ISI version) for data collection and manipulation provided by Infrasoft International (ISI, Port Matilda, PA, USA). An existing NIRS calibration for the fiber and protein fractions was refined. Therefore, the calibration and validation subsets were relatively small, with 36 and 30 calibration samples and 14 and 20 validation samples in the first and second experimental years, respectively. The subsets used for starch and fat determina-

tions consisted of 62 and 40 calibration samples and 20 validation samples in the first and second experimental years, respectively. Calibrations were later developed by regressing the laboratory-determined values of the sample subsets against the NIR spectral data. The correlations obtained between the reference analysis and the NIRS values were satisfactory. Means and ranges as well as correlation coefficients and standard errors of calibration and validation for the investigated quality parameters are presented in Table 2.

**Table 1.** Heading date (number of days after 1 April) and MSC of accessions included in the study (Hohenschulen).

| Accessions | Heading Date | MSC (First Cut) |
|---|---|---|
| 1 | 54 | 5.63 abc |
| 2 | 56 | 5.27 abcd |
| 3 | 58 | 5.01 abcd |
| 4 | 63 | 4.68 cd |
| 5 | 53 | 5.70 abc |
| 6 | 53 | 6.23 a |
| 7 | 62 | 4.23 d |
| 8 | 63 | 4.21 d |
| 9 | 61 | 4.71 cd |
| 10 | 55 | 5.41 abcd |
| 11 | 60 | 4.82 cd |
| 12 | 62 | 4.49 cd |
| 13 | 63 | 4.90 bcd |
| 14 | 55 | 5.66 abc |
| 15 | 61 | 4.76 cd |
| 16 | 64 | 4.67 cd |
| 17 | 54 | 5.57 abc |
| 18 | 55 | 4.61 cd |
| 19 | 55 | 5.24 abcd |
| 20 | 56 | 6.15 ab |
| S.E. | | 0.23 |

Means followed by different letter(s) within the same column are different at $p < 0.05$; S.E., standard error.

**Table 2.** Statistical data of NIRS calibration and validation for the different nutritive aspects of the investigated accessions.

| Parameter (g/kg DM) | n [1] | Mean | Range | $R^2$ | SEC | SEV |
|---|---|---|---|---|---|---|
| NDF | 96 | 567 | 413–693 | 0.93 | 16.00 | 20.79 |
| ADF | 92 | 280 | 174–363 | 0.97 | 7.03 | 10.28 |
| ADL | 63 | 200 | 37–411 | 0.54 | 5.60 | 5.32 |
| WSC | 96 | 98 | 16–186 | 0.93 | 9.60 | 10.30 |
| starch | 100 | 34 | 1–169 | 0.77 | 1.45 | 2.23 |
| fat | 99 | 25 | 3–45 | 0.86 | 0.32 | 0.30 |
| ash | 101 | 85 | 42–132 | 0.92 | 0.57 | 0.65 |
| DM | 65 | 922 | 897–944 | 0.95 | 0.33 | 0.75 |
| N | 99 | 20 | 5–39 | 0.98 | 0.08 | 0.08 |
| **Parameter (g N/kg N)** | | | | | | |
| protein fraction A | 123 | 261.3 | 119.4–396.7 | 0.98 | 0.87 | 1.06 |
| protein fraction B1 | 123 | 94.2 | 0.82–263.23 | 0.90 | 15.53 | 39.43 |
| protein fraction B3 | 77 | 301.3 | 29.1–447.0 | 0.82 | 36.24 | 46.98 |
| protein fraction C | 107 | 35.8 | 10.3–91.0 | 0.97 | 3.05 | 15.22 |

[1] n: number of samples; SEC: standard error of calibration; SEV: standard error of validation; NDF: neutral detergent fiber; ADF: acid detergent fiber; ADL: acid detergent lignin; WSC: water-soluble carbohydrates; DM: dry matter.

Neutral detergent fiber (NDF) and acid detergent fiber (ADF) contents were determined sequentially using the ANKOM-220 semi-automatic fiber analyzer (ANKOM Technology, Macedon, NY, USA) as described previously [26]. NDF and ADF were analyzed

without heat-stable amylase and expressed including residual ash. In contrast, ADForg indicates correction for residual ash content. The crude lipid (CL) content of the samples was determined by the traditional Soxhlet ether extraction method. Ash was determined by burning the subsample in a muffle oven at 550 °C for 3 h.

Water-soluble carbohydrate (WSC) content was measured by high-performance anion-exchange chromatography with pulsed amperometric detection, HPAEC-PAD (ICS-2500, Dionex Corp., Sunnyvale, CA, USA), with modifications [27,28]. Prior to analysis, the dried samples were reground to a particle size of 10 μm using a ball mill. To extract the WSC, 20 mg of the ground material was stirred with 2 mL of deionized cold water for 10 min. The samples were then replaced in a boiling water bath at 100 °C for 10 min. After centrifugation ($1200 \times g$) for 30 min, the supernatant was cleared with 67 μL of chloroform. The supernatant was separated from the pellet and then diluted 1:5 in deionized water, and 2 mL of the dilution was filtrated through a C18 cartridge (Strata C18-E, Phenomenex Inc., Torrance, CA, USA). The filtrate was hydrolyzed in 50 μL 2N HCl for 2 h at 80 °C to cleave fructans into glucose and fructose.

The nitrogen content was analyzed using a C/N analyzer (Vario Max CN, Elementar Analyse-Systeme, Hanau, Germany), and the crude protein (CP) content was calculated from the N content (CP = N × 6.25). Prior to starch analyses, the dried samples were again ground to a particle size of 10 μm using a ball mill. For the determination of starch content, the WSC content of the sample was removed by stirring 20 mg of the ground material with deionized cold water for 10 min and then placing the samples in a boiling water bath at 100 °C for another 10 min. After centrifugation ($1200 \times g$) for 30 min, the supernatant was separated from the pellet. The pellet was then incubated with sodium acetate buffer (100 mM, pH 4.5) and amyloglucosidase enzyme (Roche, Mannheim, Germany) at 37 °C overnight to cleave starch into glucose units. After centrifugation ($1200 \times g$) for 10 min, the supernatant was purified with 67 μL of chloroform. The supernatant was separated from the pellet and then diluted 1:1 in deionized water, and 1 mL of the dilution was filtered through a C18 cartridge (Strata C18-E, Phenomenex Inc., Torrance, CA, USA). Glucose was measured by high-performance anion-exchange chromatography with pulsed amperometric detection, HPAEC-PAD (ICS-2500, Dionex Corp., Sunnyvale, CA, USA).

### 2.5. Carbohydrate Fractions of Forages with CNCPS

Using all the previously determined nutrient components, the total carbohydrate (CHO) content, expressed as a percentage of dry matter (DM), and the carbohydrate fractions A, B1, B2, and C, expressed as a percentage of CHO, were calculated according to the formulas described by CNCPS [9]. In the CNCPS, carbohydrates are classified according to their rate of degradation: fraction A is fast and is sugars; fraction B1 is intermediate and is starch; fraction B2 is slow and is available cell wall; and fraction C is unavailable cell wall.

### 2.6. Dietary Protein Fractions Using CNCPS

Feed proteins were separated into five protein fractions, relative to total N, using the procedures as described [29]. Fraction A, representing non-protein nitrogen (NPN), was measured by precipitation of total protein with tungstic acid followed by filtration through Whatman #54 filter paper. The insoluble nitrogen in the residue was then determined. Finally, fraction A was calculated by subtracting the N content of the residue from the total N content of the sample. The true protein content, known as fraction B, is further divided into three subfractions (B1, B2, and B3) based on their inherent rates of ruminal degradation. Fraction B1 is rapidly degraded, fraction B2 is intermediately degraded, while fraction B3 represents the slowly degradable true protein. Fraction B1 was measured after placing the sample in borate–phosphate buffer (pH 6.7–6.8) followed by filtration through Whatman #541 filter paper. The N content of the residue includes the insoluble protein fraction. B1 is calculated by subtracting the insoluble protein and NPN contents from the total N content of the forage. Fraction B2 was calculated as the difference between total N and the sum

of fractions A, B1, B3, and C. To measure fractions B3 and C, neutral detergent insoluble protein (NDIN) and acid detergent insoluble protein (ADIN) were analyzed sequentially after the analysis of NDF and ADF [26]. Fraction B3, representing the nitrogen associated with NDF, was calculated by subtracting NDIN from the ADIN content. The ruminal and intestinal unavailable protein is combined in fraction C, which contains the acid detergent insoluble nitrogen (ADIN).

*2.7. Calculations and Statistical Procedures*

The metabolizable energy (ME) content was determined according to the pepsin cellulase method [30], where the enzymatically soluble organic matter (ESOM) is measured. To estimate the ME content [21], the equation applies:

$$\text{ME (MJ/kg DM)} = 5.51 + 0.00828 \times \text{ESOM} - 0.00511 \times \text{CA} + 0.02507 \times \text{CL} - 0.00392 \times \text{ADF}_{\text{org}}, \tag{1}$$

where CA is crude ash, ESOM is enzymatically soluble organic matter, CL is crude lipid, and $\text{ADF}_{\text{org}}$ is ash-free ADF content. All parameters are expressed as g/kg DM.

The content of utilizable crude protein at the duodenum (uCP) was calculated as follows [31]:

$$\text{uCP (g/kg DM)} = (11.93 - (6.82 \times (\text{UDP/CP}))) \times \text{ME} + (1.03 \times \text{UDP}), \tag{2}$$

where ME is metabolizable energy in MJ/kg DM, CP is crude protein, and UDP is undegradable dietary protein in the rumen, both expressed in g/kg DM. uCP is considered as the sum of UDP and the amount of microbial protein synthesized in the rumen, both representing the main source of amino acids for the host.

Data from both years were included in the statistical model, but the year was not further considered as the year effect is confounded with sward age. The trial location and accession and their interaction were tested for significance using Proc Mixed of, [®]9.1 (SAS Institute, Inc., Cary, NC, USA [32]). Only replicates were considered random. The nutritional parameter was then analyzed according to the following model:

$$Y_{ijk} = \mu + S_i + G_j + R_k + (S \times G)_{ij} + e_{ijk}, \tag{3}$$

where $\mu$ is the general mean, $S_i$ is the site effect (i = 1, 2, 3), $G_j$ is the accession effect (j = 1–20), $R_k$ is the replication (k = 1, 2, 3), and $e_{ijk}$ is the residual error. The first cut and mean annual quality data are shown. In both experiments, significance was declared at $p < 0.05$, means with significant F-values were tested with Student's *t*-test, and probabilities were corrected by the Tukey–Kramer test.

## 3. Results

The difference in heading dates between the earliest and latest accessions was 11 days, which suggests that these differences were sufficient to exert a substantial influence on the nutritive value of the accessions. In addition, the MSC data confirm that the 20 tested accessions displayed different maturation behaviors at the time of the first cut (Table 1).

Table 3 shows the statistics of all forage quality parameters. The results for the content of NDF, ADF, and WSC are shown in Table 4. Accession 9 has the lowest NDF and ADF content, whereas accessions 10 and 18 have the highest WSC content. In contrast, accession 6 has the lowest WSC content, as expected, but the highest NDF and ADF content.

*Ruminal Kinetics*

Analyses of the first cut (Table 5) and annual (Table 6) ruminally digested and undigested carbohydrates and proteins showed variation among the three sites and the 20 accessions. In addition, a 'site × accession' interaction was found for the annual means (Table 3 and means in Table 7).

**Table 3.** Overview of statistical effects of site, accession, and their interaction on first cut and annual yield, carbohydrate (g/kg CHO) and protein (g/kg N) fractions, total carbohydrate and total protein contents (g/kg DM), as well as ruminally digested (RDC) and undigested (UDC) carbohydrate (g/kg CHO) contents and ruminally degraded (RDP) and undegraded (UDP) protein (g/kg N) contents over the two experimental years.

| | | First Cut | | | Annual Yield | | |
|---|---|---|---|---|---|---|---|
| | | **Site (S)** | **Accession (G)** | **S × G** | **Site (S)** | **Accession (G)** | **S × G** |
| Carbohydrate fractions | A | *** | *** | ns | ** | *** | * |
| | B1 | *** | ns | ns | *** | *** | *** |
| | B2 | *** | *** | ns | *** | *** | * |
| | C | *** | *** | ns | ns | *** | ** |
| Protein Fractions | A | *** | ** | ns | *** | ns | ns |
| | B1 | *** | *** | ns | *** | *** | * |
| | B2 | ** | *** | ns | ns | *** | ns |
| | B3 | *** | *** | ns | ** | *** | ns |
| | C | *** | *** | ns | ** | *** | ns |
| Total CHO | | *** | *** | ns | ** | *** | ns |
| Total N | | *** | *** | *** | *** | *** | * |
| Total RDC | | *** | *** | ns | * | *** | * |
| Total RDP | | *** | *** | ns | *** | *** | * |
| Total UDC | | *** | *** | ns | * | *** | * |
| Total UDP | | *** | *** | ns | *** | *** | * |

\* $p < 0.05$; \*\* $p < 0.01$; \*\*\* $p < 0.001$; ns $p > 0.05$.

**Table 4.** LS means of first cut NDF, ADF, and WSC (g/kg DM) of the 20 tested accessions over 2006 and 2007.

| Accessions | NDF | ADF | WSC |
|---|---|---|---|
| 1 | 530.1 bcde | 270.4 bcd | 125.8 ab |
| 2 | 525.9 bcde | 269.0 bcd | 130.0 ab |
| 3 | 519.6 bcdefg | 257.9 de | 127.3 ab |
| 4 | 500.4 gh | 241.8 f | 134.6 ab |
| 5 | 536.8 ab | 275.9 abc | 126.5 ab |
| 6 | 553.3 a | 287.0 a | 122.8 b |
| 7 | 518.5 bcdefg | 251.4 ef | 128.1 ab |
| 8 | 516.6 cdefgh | 252.1 ef | 128.7 ab |
| 9 | 498.4 h | 242.0 f | 131.6 ab |
| 10 | 503.6 fgh | 250.2 ef | 137.8 a |
| 11 | 520.9 bcdef | 260.5 cde | 128.0 ab |
| 12 | 519.3 bcdefg | 257.7 def | 126.7 ab |
| 13 | 511.2 efgh | 256.9 def | 129.3 ab |
| 14 | 534.7 abc | 274.1 abc | 128.5 ab |
| 15 | 531.8 bcd | 268.7 bcd | 127.2 ab |
| 16 | 511.7 efgh | 250.9 ef | 129.3 ab |
| 17 | 529.8 bcde | 265.3 bcde | 128.2 ab |
| 18 | 514.6 defgh | 249.8 ef | 136.6 a |
| 19 | 520.6 bcdef | 263.9 bcde | 132.0 ab |
| 20 | 538.2 ab | 276.9 ab | 124.8 ab |
| S.E. | 4.0 | 3.2 | 25.2 |

Means followed by different letter(s) within the same column are different at $p < 0.05$; S.E. standard error.

In Table 5, accession 6 was characterized by its high RDP (721.3 g/kg N) and UDC (333.4 g/kg CHO) contents in the first cut, accompanied by the lowest RDC (666.6 g/kg CHO) and UDP (278.7 g/kg N) contents among all tested accessions. The highest RDC component was in favor of the high-sugar accession 9 (706.2 g/kg CHO); it was

also characterized by its moderate RDP and UDP contents, amounting to 709.1 and 290.9 g/kg N, respectively.

**Table 5.** LS means of first cut ruminally digested (RDC) and undigested (UDC) carbohydrate (g/kg CHO) contents and ruminally degraded (RDP) and undegraded (UDP) protein (g/kg N) contents of the 20 tested accessions.

| Accessions | RDC | RDP | UDC | UDP |
|---|---|---|---|---|
| 1 | 680.8 defg | 721.5 a | 319.2 abcd | 278.5 c |
| 2 | 686.1 cdef | 718.5 abc | 313.9 bcde | 281.5 abc |
| 3 | 686.6 bcdef | 714.0 abc | 313.4 bcdef | 286.0 abc |
| 4 | 699.9 abc | 707.1 c | 300.1 efg | 293.0 a |
| 5 | 677.6 efg | 716.1 abc | 322.4 abc | 283.9 abc |
| 6 | 666.6 g | 721.3 a | 333.4 a | 278.7 c |
| 7 | 684.2 cdef | 710.8 abc | 315.8 bcde | 289.2 abc |
| 8 | 688.0 bcdef | 706.7 c | 312.0 bcdef | 293.3 a |
| 9 | 706.2 a | 709.1 abc | 293.8 g | 290.9 abc |
| 10 | 703.3 ab | 717.6 abc | 296.7 fg | 282.4 abc |
| 11 | 685.7 cdef | 715.2 abc | 314.2 bcde | 284.8 abc |
| 12 | 687.5 bcdef | 708.1 bc | 312.5 bcdef | 292.0 ab |
| 13 | 695.5 abcd | 712.1 abc | 304.5 defg | 287.9 abc |
| 14 | 680.1 defg | 713.5 abc | 319.9 abcd | 286.5 abc |
| 15 | 677.8 efg | 707.0 c | 322.2 abc | 293.0 a |
| 16 | 692.4 abcde | 706.8 c | 307.6 cdefg | 293.2 a |
| 17 | 679.2 defg | 715.4 abc | 320.7 abcd | 284.6 abc |
| 18 | 687.6 bcdef | 708.7 bc | 312.4 bcdef | 291.4 ab |
| 19 | 690.0 abcdef | 719.5 ab | 310.0 bcdefg | 280.5 bc |
| 20 | 674.5 fg | 718.2 abc | 325.5 ab | 281.9 abc |
| S.E. | 3.3 | 2.8 | 3.3 | 2.8 |

Means followed by different letter(s) within the same column are different at $p < 0.05$; S.E., standard error.

**Table 6.** LS means of annual ruminally digested (RDC) carbohydrate (g/kg CHO) and ruminally degraded (RDP) protein (g/kg N) contents of the 20 tested accessions.

| Accessions | RDC | | | RDP | | |
|---|---|---|---|---|---|---|
| | Asendorf | Malchow | Hohenschulen | Asendorf | Malchow | Hohenschulen |
| 1 | 657.9 bcd | 664.4 abc | 650.5 bcde | 691.3 a | 692.2 a | 716.7 a |
| 2 | 662.5 abcd | 661.0 abc | 666.1 abc | 689.1 a | 692.2 a | 712.3 ab |
| 3 | 659.2 bcd | 660.6 abc | 656.2 abcd | 694.2 a | 686.6 a | 705.4 ab |
| 4 | 673.6 abc | 669.1 abc | 663.1 abc | 687.0 a | 685.0 a | 707.9 ab |
| 5 | 653.8 cd | 654.3 c | 660.8 abcde | 684.2 a | 683.3 a | 709.1 ab |
| 6 | 646.8 d | 651.2 c | 651.9 abcde | 695.6 a | 690.7 a | 707.1 ab |
| 7 | 662.3 abcd | 657.1 bc | 637.8 e | 687.2 a | 683.4 a | 699.9 b |
| 8 | 663.2 abcd | 660.1 abc | 654.0 abcde | 689.2 a | 679.3 a | 698.1 b |
| 9 | 685.3 a | 678.3 ab | 670.3 ab | 687.2 a | 681.7 a | 700.9 ab |
| 10 | 680.3 ab | 682.0 a | 674.6 a | 682.7 a | 690.0 a | 706.7 ab |
| 11 | 657.4 bcd | 668.1 abc | 648.0 bcde | 685.4 a | 685.4 a | 706.4 ab |
| 12 | 661.5 bcd | 663.6 abc | 662.9 abcd | 683.1 a | 682.3 a | 703.2 ab |
| 13 | 676.3 abc | 668.0 abc | 664.3 abc | 689.9 a | 686.0 a | 711.8 ab |
| 14 | 660.7 bcd | 661.5 abc | 655.4 abcde | 690.6 a | 682.0 a | 707.8 ab |
| 15 | 656.6 cd | 662.6 abc | 644.0 cde | 685.0 a | 687.7 a | 713.4 ab |
| 16 | 667.2 abcd | 667.8 abc | 655.0 abcde | 685.9 a | 684.5 a | 707.9 ab |
| 17 | 661.6 abcd | 662.2 abc | 649.4 bcde | 687.3 a | 688.0 a | 710.5 ab |
| 18 | 663.4 abcd | 659.3 abc | 639.3 de | 686.2 a | 689.7 a | 708.8 ab |
| 19 | 670.1 abcd | 666.9 abc | 670.1 ab | 690.4 a | 690.5 a | 704.7 ab |
| 20 | 656.6 cd | 664.3 abc | 653.9 abcde | 685.2 a | 695.5 a | 716.8 a |
| S.E. | 4.1 | | | 3.5 | | |

Means followed by different letter(s) within the same column are different at $p < 0.05$; S.E., standard error.

A similar trend was observed for the annual mean data (Tables 6 and 7). Accession 6 had the lowest amount of RDC and the highest amount of UDC in Asendorf and Malchow. In Hohenschulen, however, the ranking was slightly shifted, where accession 7 had the

lowest RDCs and the highest UDCs. The highest amount of carbohydrates digested in the rumen occurred in the high-sugar accessions 9 and 10 in the three locations. A slight variation was also observed in the difference between the highest and lowest accessions among the three sites for RDC and UDC. In general, the average highest RDC content across the three sites was 680.6 g/kg CHO, while the average lowest was 654.3 g/kg CHO.

**Table 7.** Means of annual ruminally undigested (UDC) carbohydrate (g/kg CHO) and ruminally undegraded (UDP) protein (g/kg N) contents of the 20 tested accessions.

| Accession | UDC | | | UDP | | |
|---|---|---|---|---|---|---|
| | Asendorf | Malchow | Hohenschulen | Asendorf | Malchow | Hohenschulen |
| 1 | 342.1 abcd | 335.6 abc | 349.5 abc | 308.7 a | 307.8 a | 283.3 b |
| 2 | 337.5 bcd | 339.0 abc | 333.9 bcde | 310.9 a | 307.8 a | 287.7 ab |
| 3 | 340.8 abcd | 339.4 abc | 343.8 abcde | 305.8 a | 313.4 a | 294.6 ab |
| 4 | 326.4 cd | 330.9 bc | 336.9 bcde | 313.0 a | 315.0 a | 292.1 ab |
| 5 | 346.2 ab | 345.7 ab | 339.2 abcde | 315.8 a | 316.7 a | 290.9 ab |
| 6 | 353.3 a | 348.8 a | 348.1 abcd | 304.5 a | 309.3 a | 292.9 ab |
| 7 | 337.7 abcd | 343.0 abc | 362.2 a | 312.8 a | 316.6 a | 300.2 a |
| 8 | 336.8 bcd | 339.9 abc | 346.0 abcde | 310.9 a | 320.7 a | 301.9 a |
| 9 | 314.7 d | 321.7 c | 329.7 de | 312.8 a | 318.3 a | 299.1 ab |
| 10 | 319.7 cd | 318.0 c | 325.5 e | 317.3 a | 310.0 a | 293.4 ab |
| 11 | 342.6 abcd | 331.9 abc | 352.0 abc | 314.6 a | 314.7 a | 293.6 ab |
| 12 | 338.5 abcd | 336.4 abc | 337.1 abcde | 316.9 a | 317.7 a | 296.9 ab |
| 13 | 323.8 cd | 332.0 abc | 335.7 bcde | 310.1 a | 314.0 a | 288.2 ab |
| 14 | 339.3 abcd | 338.5 abc | 344.6 abcde | 309.4 a | 318.0 a | 292.2 ab |
| 15 | 343.4 abc | 337.4 abc | 356.0 ab | 315.0 a | 312.3 a | 286.7 ab |
| 16 | 332.8 bcd | 332.2 abc | 345.0 abcde | 314.1 a | 315.5 a | 292.1 ab |
| 17 | 338.4 abcd | 337.8 abc | 350.6 abc | 312.7 a | 312.0 a | 289.5 ab |
| 18 | 336.2 bcd | 340.7 abc | 360.7 ab | 313.8 a | 310.3 a | 291.2 ab |
| 19 | 329.9 bcd | 333.1 abc | 329.9 cde | 309.6 a | 309.5 a | 295.3 ab |
| 20 | 343.4 abc | 335.7 abc | 346.2 abcde | 314.8 a | 304.5 a | 283.2 b |
| S.E. | | 4.1 | | | 3.5 | |

Means followed by different letter(s) within the same column are different at *p* < 0.05; S.E., standard error.

A different trend to the carbohydrate kinetics was obtained from the protein kinetics, represented by the RDP and UDP proportions (Tables 6 and 7). Regarding the significant two-way interaction (Table 3), significant differences among the tested accessions for RDP and UDP proportions were observed only in Hohenschulen. Accessions 1 and 20 were characterized by the highest RDP amounts (716.7 and 716.8 g/kg N, respectively) and consequently the lowest UDP amounts (283.3 and 283.2 g/kg N, respectively). Conversely, accessions 7 and 8 had the lowest RDP content (699.9 and 698.1 g/kg N, respectively) but the highest UDP content (300.2 and 301.9 g/kg N, respectively). Remarkably, the difference between the highest and lowest values in both divisions was very small and amounted to 18.7 g/kg N.

Tables 6 and 7 show the amount (in g/kg DM) of degraded and undegraded CHO and CP as measured in this study. Accession 6 showed a lower proportion of RDC and a higher proportion of UDC, while the content of RDP and UDP remained unchanged among them.

Table 8 shows the potential milk production based on the amount of uCP reaching the duodenum using Equation (2) for accessions 6 and 9.

**Table 8.** Estimation of potential milk production by the amount of uCP [1] arriving in the duodenum.

| | in g/kg DM | Accession 6 | Accession 9 |
|---|---|---|---|
| Degraded | carbohydrate (RDC) | 538.1 b | 562.4 a |
| | protein (RDP) | 11.4 a | 12.2 a |
| Undegraded | carbohydrate (UDC) | 269.1 a | 234.0 b |
| | protein (UDP) | 4.4 a | 5.0 a |
| | uCP [1] | 134 | 145 |
| Calculated milk production from uCP, kg/day | | 15.4 | 17.0 |

[1] uCP: usable crude protein at the duodenum. Means followed by different letter(s) within the same line are different at *p* < 0.05

## 4. Discussion

After measuring the MSC of all the accessions, a good correlation with the quality parameters was observed [33]. This means that the nutritional variations observed depend on the time of harvest and, at that time, on the maturity of the accessions. The maturity of a grass sward is a key factor in explaining the changes that occur in its nutritional value. To make matters worse, in most European countries, variation in maturity within the same heading group may occur, but the nutritive value is not a standard measurement in variety candidate trials. Variations in the nutritive parameters studied may be largely due to differences in maturity, as observed for crude protein and fiber content, which may change unfavorably with advancing maturity [34].

Nutrient requirement systems based on in vivo metabolism have been recognized for many years, such as the rumen modeling efforts [35,36]. These models were complex, mechanistic, and dynamic, originating at the biochemical level. The Weender proximate analysis system and the total digestible nutrient system have been used for more than a century as the basis for predicting the energy and protein available from forages in ruminant diets [37]. Later, net energy (NE) systems were developed. The NE systems adjust requirements for methane, urine, and heat increment losses [38–40]. However, they were not accurate enough to account for the variable feeding conditions of cattle [41]. In addition, these parameters did not provide sufficiently detailed information to be used as selection targets in forage plant breeding.

The use of CNCPS to evaluate forages for breeding purposes is of interest to animal nutritionists and plant breeders [42,43]. Total nonstructural carbohydrates in forages provide a source of rapidly available energy in the rumen, and increased concentrations improve the N use efficiency of dairy cows [8,44]. Variation was observed among the 20 accessions tested in the carbohydrate and protein fractions examined (Tables S1 and S2). However, the degree of variability was small. The differences between the highest and lowest values were about 6, 1, 5, and 2% CHO points for the carbohydrate fractions A, B1, B2, and C, respectively. Assuming that fraction A contains WSC, a difference of 6% was observed for the first cut for the high-sugar accession (9) and the low-sugar accession (6), which is higher or comparable to mass fractions found in other studies [8,45,46]. The difference was smaller for the protein fractions, accounting for approximately 4, 3, 4, 3, and 0.5% N differences between accessions 6 and 9 for the first cut fractions A, B1, B2, B3, and C, respectively. In addition to the limited variability of the protein fractions, the amounts determined were generally small compared to similar studies [45]. Almost a quarter of the total protein was in the form of non-protein nitrogen (fraction A), the true protein is mainly composed of fractions B1 and B2, with a small proportion of fraction B3, and the protein fraction C represents the smallest part of the first cut total N. Such observations are consistent with others [24,42,43,45,47,48].

The CP content of our tested accessions was low (Table 2, average N content is 20 g/kg DM, i.e., 125 g CP/kg DM). CP fractionation should be performed also in accessions with higher CP content, as the N fertilizer application rate may have a diverse effect on CP content [49] and CP fractions. The negative effect of higher N application rate on CHO fractions, especially RDC content, is uncertain [45,47]. Changes in RDC content in our study are associated with lower incorporation of WSC into complex carbohydrates (high RDC content and lower UDC in accession 9, Tables 6 and 7). These results demonstrate the breeding progress. In contrast, RDP and UDP did not change [50]. This result is consistent with the findings of others for perennial ryegrass [8] and for timothy [51]. It confirms that the negative correlation between WSC and N does not hold when comparing different genetic lines [46].

Despite the small variation found among the 20 accessions tested, deeper insights into the kinetics of the different carbohydrate and protein fractions in the rumen, which directly affect their rate of degradation in the rumen, were obtained. Using accessions 6 and 9 as examples for the conventional and high-sugar accessions, respectively, the data showed a difference in ruminally digested carbohydrates of 24 g RDC/kg DM in favor of

the high-sugar accession. Together with the similar ruminally degraded protein content for the respective accessions 6 and 9, accounting for 11 and 12 g of RDP/kg DM, it could be suggested that the high-sugar accession 9 has a better chance of achieving a balanced carbohydrate/protein metabolism in ruminants [3]. In fact, the study [52] implies that 32 g of N per kg of available carbohydrate should be the ratio of rumen-available CP, expressed as N, to rumen-available carbohydrate. Applying this suggested value to our results showed that all our tested accessions had sufficient RDC content to degrade the available RDP. As mentioned above, this could be explained by the lower protein content of the tested accessions and/or the similarity in RDC content between cultivars.

The significant improvement in the performance of meat and dairy animals when the supply of amino acids to the small intestine has been optimized is well documented [22]. Many studies have highlighted CNCPS as an efficient and reliable evaluation system for predicting the UDP content of different diets and feedstuffs [53,54]. Thus, selecting grasses for increased RDC content is an effective way to ensure that the cow is supplied with an adequate amount of amino acids from dietary carbohydrates (enhancement of microbial protein synthesis in the rumen) to support animal requirements. Calculated on a dry matter basis, the UDP content of the two respective accessions 6 and 9 was 4.4 and 5 g UDP/kg DM, which is not a relevant difference. However, in combination with their higher RDC content, accessions 6 and 9 indicate the superiority of carbohydrates, rather than crude protein and protein fractions, in supporting higher N use efficiency in ruminant diets. Increasing the RDC content, as in high-sugar grasses, may therefore support microbial growth in the rumen and efficient N use by animals. In contrast to UDP, the higher RDC would allow a higher N utilization in the rumen for microbial growth, thus reducing N losses, especially under grazing conditions. Calculations using grasses with higher RDC (Table 8) support an 8% higher proportion of protein flow to the duodenum (usable crude protein at the duodenum, [31]).

## 5. Conclusions

The results of the following study suggest that carbohydrate and protein fractionation using the CNCPS can provide accurate information on the differences between grass accessions and their potential use in the rumen. Consequently, the selection of grasses using CNCPS parameters may assist in the composition of more appropriate diets that meet the rumen microbial and animal requirements to achieve their appropriate maintenance and production levels. This is especially important in forage-based production systems. Using the CNCPS to screen the 20 perennial ryegrass accessions, it was found that high-sugar accessions, characterized by low fiber content and high digestibility, would positively contribute to improving the carbohydrate/protein balance in ruminant diets. Therefore, there is a potential to reduce the risk of high ammonia excretion and environmental pollution, as well as to avoid concentrated feeding to support high-yielding animals. Based on our findings, the analysis of parameters related to ruminal degradation kinetics should favor the selection of accessions with higher amounts of ruminally digested carbohydrates (RDC). The selection of accessions based on protein quality (proportion of UDP) is less favorable for achieving higher levels of usable protein in small intestines (sum of UDP and amount of microbial protein synthesized in the rumen). The large variation in the development stage at harvest suggests that this parameter should be included as a covariate in the statistical model when comparing the nutritive value of varieties, even for those with similar genetic backgrounds.

**Supplementary Materials:** The following supporting information can be downloaded at: https://www.mdpi.com/article/10.3390/agronomy14010168/s1, Table S1: LS-means of first cut carbohydrate and protein fractions and total carbohydrate content of the 20 tested accessions over two years. Table S2: LS-means of first cut and annual total nitrogen content of the 20 tested accessions as affected by site × accession interaction over two years.

**Author Contributions:** Conceptualization, M.G. and A.H.; methodology, H.S.A.S. and M.L.; software, H.S.A.S. and M.L.; validation, H.S.A.S. and M.L.; formal analysis, H.S.A.S. and M.L.; investigation, H.S.A.S. and M.L.; resources, H.S.A.S. and M.L.; data curation, H.S.A.S. and M.L.; writing—original draft preparation, H.S.A.S. and M.L.; writing—review and editing, M.G., A.H., H.S.A.S. and M.L.; visualization, M.G., A.H., H.S.A.S. and M.L.; supervision, M.G., A.H. and F.T.; project administration, A.H. and F.T.; funding acquisition, M.G., A.H. and F.T. All authors have read and agreed to the published version of the manuscript.

**Funding:** This research was financially supported by the German Academic Exchange Service (DAAD) and the Innovationsstiftung Schleswig-Holstein (Project Nr. 205-22H).

**Data Availability Statement:** The data presented in this study are available on request from the corresponding author.

**Conflicts of Interest:** The authors declare no conflicts of interest.

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
