# Peer review of "Protein and Carbohydrate Fractionation to Evaluate Perennial Ryegrass (Lolium perenne L.) Accessions"

_agronomy, doi:10.3390/agronomy14010168_

Round 1

Reviewer 1 Report

Comments and Suggestions for Authors

The authors presented a very interesting approach to the problem of ruminal kinetics of carbohydrate and protein fractions and using it in breeding new varieties of grass. All sections of the manuscript are written correctly and the results are well documented.

However, as a reviewer, I have the following comments:

1. The authors should explain what the accession selection criteria were for the study.

2. The CP content was actually relatively low (average 125 g CP/kg DM). However, I do not agree with the explanation that it resulted from the use of 300 kg N/ha (by the way in the line 365 is 300 g N/ha!). The dose of 300 kg N/ha is a high dose, currently not used in agricultural practice.

3. The Discussion should include a sentence stating that it would be worth conducting tests for plant samples with a higher CP content.

4. Line 90; Lee et al. 2001 is not mentioned in the References

5. Line 109; is 80.2 should be 802 (I think)

6. Line 160 and 161; what does it mean “AD-Forg”?

7. Table 2; the abbreviation ADL is used only once throughout the text and should be explained; WSC is used many times but there is also no explanation of this abbreviation

8. Line 170; Chatterton et al. (1989) and Shiomi et al. (1991) are not mentioned in the References

9. Line 260; is “accession 10 has the highest WSC content”, should be “accessions 10 and 18 have the highest WSC content”

10. Line 274; I don't understand why the Authors refer to Table 3

11. In the Tables 6 and 7 are given data from three locations. What about the Table 5? Are there means from three locations?

Author Response

A: We would like to express our sincere gratitude for your thorough review of our manuscript. Your insightful comments and suggestions have helped to improve the quality and clarity of our work.

The authors presented a very interesting approach to the problem of ruminal kinetics of carbohydrate and protein fractions and using it in breeding new varieties of grass. All sections of the manuscript are written correctly and the results are well documented.

However, as a reviewer, I have the following comments:

  1. The authors should explain what the accession selection criteria were for the study.

A: The answer is in line 111-112: The plant material studied consisted of twenty diploid perennial ryegrass (Lolium perenne L.) accessions with intermediate heading.

  1. The CP content was actually relatively low (average 125 g CP/kg DM). However, I do not agree with the explanation that it resulted from the use of 300 kg N/ha (by the way in the line 365 is 300 g N/ha!). The dose of 300 kg N/ha is a high dose, currently not used in agricultural practice.

A: we agree and delete the explanatory sentence.

  1. The Discussion should include a sentence stating that it would be worth conducting tests for plant samples with a higher CP content.

A: done

  1. Line 90; Lee et al. 2001 is not mentioned in the References

A: included

  1. Line 109; is 80.2 should be 802 (I think)

A: corrected, thank you!

  1. Line 160 and 161; what does it mean “AD-Forg”?

A: corrected

  1. Table 2; the abbreviation ADL is used only once throughout the text and should be explained; WSC is used many times but there is also no explanation of this abbreviation

A: corrected

  1. Line 170; Chatterton et al. (1989) and Shiomi et al. (1991) are not mentioned in the References

A: included

  1. Line 260; is “accession 10 has the highest WSC content”, should be “accessions 10 and 18 have the highest WSC content”

A: corrected

  1. Line 274; I don't understand why the Authors refer to Table 3

A: the Table 3 was erroneously mentioned here - deleted

  1. In the Tables 6 and 7 are given data from three locations. What about the Table 5? Are there means from three locations?

A: As shown in Table 3, only for the annual averages the interaction Site x Genotypes was significant. Table 5 shows the results for the first cut.

Reviewer 2 Report

Comments and Suggestions for Authors

Please see the attached file for review comments.

Author Response

A: We would like to express our sincere gratitude for your thorough review of our manuscript. Your insightful comments and suggestions have helped to improve the quality and clarity of our work.

In the manuscript “Using parameters of ruminant nutrition to differentiate perennial ryegrass (Lolium perenne L.) accessions” the authors suggest that carbohydrate and protein fractionation using the Cornell Net Carbohydrate and Protein System can provide accurate information on the differences between grass accessions in relation to their utilization in the rumen. This information is important in forage-based production systems.

Although the subject is interesting and methods chosen for the study are appropriate, some important concerns need to be addressed before the manuscript is ready for publication in Agronomy.

General comments:

  1. The presentation of results should be

A: done

  1. References and their citations should be carefully verified. Not all references cited in the text are listed in the References section. Authors names should be provided in a consistent manner.

A: verified and corrected

Specific comments:

  1. Line 50: The abbreviations UDP and CP should be defined. Acronyms/Abbreviations should be defined the first time they appear in each of three sections: the abstract; the main text; the figure or table.

A: done

  1. Lines 85-86: See comment 3 for UDP.

A: done

  1. Lines 90-91: The reference Lee et al. 2001 is not listed in the References This important issue should be addressed.

A: done

  1. Line 96: See comment 3 for CP.

A: done

  1. Line 139, Table 1 title: It should be “Heading date (number of days after April 1st) and MSC of accessions” instead of “Heading date (number of days after April 1st and MSC of accessions”.

A: done

  1. Line 167, Table 2 footnotes: I suggest to provide definitions for the abbreviations NDF, ADF, ADL, WSC, DM (see comment no. 3).

A: done

  1. Line 168: I suggest “The water-soluble carbohydrate (WSC) content was measured“ or “The WSC content was measured” instead of “WSC was measured” (see comments no. 8 and 3 for comparison).

A: done

  1. Line 170: The references Chatterton et (1989) and Shiomi et al. (1991) are not listed in the

References section. This important issue should be addressed.

A: Done

  1. Line 174: I suggest to describe centrifugation conditions in terms of g force (see line 185 for comparison).

A: done

  1. Line 187: Sodium acetate buffer should be more precisely described (molarity and pH).

A: done

  1. Line 205: I suggest to avoid using abbreviations if they occur only once in the text (TP).

A: done

  1. Line 212: I suggest to describe borate-phosphate buffer more precisely (molarity and pH).

A: Details about the procedures are described by Licitra et al. (1996).

  1. Line 226: ME should be expressed in “MJ/kg DM” instead of “MJ kg/DM”.

A: corrected

  1. Line 228: I suggest “crude lipid” instead of “crude fat” for the abbreviation “CL”.

A: done

  1. Line 229: It should be “g kg-1 DM” instead of “g kg-1 DM”. However, I suggest using “g/kg DM” (see line 234 for comparison).

A: done

  1. Line 231: An unnecessary yellow

A: done

  1. Line 232: Two necessary spaces are missing next to the equal

A: done

  1. Line 234: I suggest “uCP” instead of ”UCP”.

A: done

  1. Line 235: The statement “the sum of UDP and rumen microbial protein synthesis” should be corrected. See lines 430-431 for comparison.

A: done

  1. Lines 237-242: Table 3 should be moved from the Materials and Methods section to the Results

section. See also line 258 for comparison.

A: done

  1. Line 247: I suggest to avoid using abbreviations if they occur only once in the text (NP).

A: done

  1. Lines 272-274: The sentence “In Table 5, accession 6 was characterized by its high RDP (721.3 g/kg N) and UDC (333.4 g/kg CHO) contents in the first cut, accompanied by the lowest RDC (666.6 g/kg CHO) and UDP (278.7 g/kg N) contents among all tested accessions (Table 3).” is difficult to understand because of “Table 3”.

A: corrected

  1. Lines 302-306: The statements regarding Table 6 are incorrect. The results described in lines 302- 306 are presented in Table 8.

A: done

  1. Line 332: It should be “(Simon and Park 1983)” instead of “(Simon and Park, 1983)”. See lines 41, 44-45, 47-48, 63, 375 for comparison.

A: done

  1. Line 336: I suggest to avoid using abbreviations if they occur only once in the text (TDN).

A: done

  1. Lines 364-365: The sentence “The CP content of our tested accessions was low (Table 1, average N content is 20 g/kg DM, i.e. 125 g CP/kg DM).” is difficult to understand because of “Table 1”.

A: done

  1. Lines 370-372: In addition to Table 6, Table 7 should also be mentioned due to the UDC

A: done

  1. Lines 373-375: The sentence “In combination with a higher utilization of carbon for protein synthesis and for the production of energy required for nitrate reduction, a step preceding protein synthesis (Reid and Strachan 1974), which could be an explanation” is difficult to understand.

A: done

  1. Lines 394-395: Table 8 should be moved from the Discussion section to the Results See also comment no. 25.

A: done

  1. Line 405: I suggest “4.4 and 5 g UDP/kg DM, which is not a relevant difference” instead of “4 and 5 g UDP/kg DM, which is not a relevant difference”.

A: done

  1. Lines 411-412: I suggest “Calculations using grasses with higher RDC (Table 8) support” instead of “Calculations using grasses (Table 8) with higher RDC support”

A: done

  1. Lines 435-436: The Supplementary Materials section should be corrected. Tables S1 and S2 are

A: done

  1. Table S2 is not mentioned in the This important issue should be addressed.

A: done

  1. Lines 445-449: The Data Availability Statement section should be

A: done

  1. The References section: The list of references should be extended (see comments 5 and 10).

A: done

  1. The References section: References should be listed in an alphabetic order – see 4 and Ref. 9.

A: done

  1. The References section: Authors names should be provided in a consistent manner - see Refs. 2, 3, 12-14, 19, 20, 36, 39.

A: done
